# OpenReview forum: "Anti-Backdoor Coreset Selection via the Cumulative Entropy Criterion"
_ICLR.cc/2026/Conference — Submitted to ICLR 2026_

### Official Review · Reviewer_ERGX · 2025-10-22

**Soundness:** 3
**Presentation:** 3
**Contribution:** 2
**Rating:** 4
**Confidence:** 2

**Summary:**

The paper presents a training-time backdoor defence framed as a coreset-selection recipe built on a rigid assumption: poisoned samples follow different learning dynamics; They become confidently classified much earlier than benign ones.
It uses cumulative entropy over epochs, with per-epoch normalisation, to rank and keep the “informative” subset, followed by warm-up + unlearning + retrain.
Because this assumption holds mainly for non-adaptive, simpler attacks, the method performs well on standard benchmarks but weakens on adaptive or stealthy attacks.

**Strengths:**

- The paper packages known ingredients, coreset selection, cumulative learning-dynamics signals, and light unlearning, into a tidy training-time pipeline for backdoor defence.
- The proposal is methodological, and the empirical results are positive across diverse attacks/datasets.
- It is a convincing engineering package of some established ideas.

**Weaknesses:**

- The novelty is in integration and execution rather than fundamentally new mechanisms. As mentioned in the paper, learning dynamics, coreset, entropy, and cross-epoch aggregation are priorly studied ingredients for the proposed pipeline.
- The key assumption that poisons learn fast and become confident early may not hold in adaptive attacks. It would be helpful to include an experiment with an adaptive attack targeting this assumption.
- The assumption of the lack of reference/validation data is also overly strong. It would be helpful to include baselines that use reasonable amount of reference data.
- There are quite a few hyperparameters whose robustness should be tested.
- With the coreset retained being considerably smaller than the original data, it is unclear whether rare and complex cases (particularly in imbalanced problems) would be discarded.

**Questions:**

1. What is the theoretical motivation for the min-max scaling of entropy? How does the range of changes change as training progresses?
2. What is the difference between aggregated entropy and AUC (under the entropy curve)?
3. Since you train on a subset, can you report tail/edge-case accuracy?

---

> ### Author Response · Authors · 2025-11-20
> **Reply to reviewer ERGX (1)**
>
> We thank the reviewer for their thoughtful feedback. Our method's novelty indeed lies in the integration and execution of existing concepts BUT to introduce a novel defense paradigm that is highly practical and effective. We remove **the need for perfect dataset separation** and **the reliance on pre-identified poisoned samples**. Prior defenses discard too many benign samples or fail to fully remove poisoned ones, leading to degraded natural accuracy or incomplete defense (as shown in Table 1).
>
> In following, we address the mentioned weakness and answer all questions, individually:
>
> **W2: The assumption of easy-learning for backdoors might not hold in an adaptive attack.**
>
> Training on datasets with noisy labels has been shown to increase learning difficulty [1], inducing higher predictive uncertainty [2, 3].
> A straightforward approach is to introduce label randomization by randomly reassigning the labels of a subset of poisoned samples.
> This increases the learning difficulty of poisoned samples and hinders backdoor memorization, thus slowing down its convergence (cf. **Appendix Figure 17**).
>
> Concretely, for a poisoning ratio of 5%, increasing random-labeling ratio (percentage of poisoned samples with randomized labels) significantly weakens the attack, as reflected by a lower ASR at higher label randomization levels.
> Additionally, we summarize the corresponding experimental results across different attacks in **Appendix D.8, Table 14**.
>
> In summary: A high level of randomized labels (e.g., 50% for BadNets) renders the backdoor substantially less effective.
> At the same time, our defense exhibits limited suppression in this setting, primarily because excessive randomization makes it difficult to isolate sufficient poisoned samples.
> Nevertheless, the defense effectiveness correlates with the backdoor strength.
>
> For instance, in WaNet with a 50% random-labeling ratio, the attack remains strong (ASR ≈ 78%), while our defense correspondingly becomes more effective.
> Similarly, as the random-labeling ratio decreases and the backdoor attack becomes stronger, and our defense reduces ASR more substantial, demonstrating its effectiveness across different attack intensities.
>
> **W3: The assumption of lacking reference data is too strong.**
>
> The absence of any reference dataset is a strong, practical constraint that indeed makes defense difficult.
> To highlight this benefit of ABCS even more apparent, we include the "Adaptively Splitting Defense" (ASD) [4] as an additional baseline.
> ASD has access to a small amount of clean reference data, constructed by randomly sampling 10 clean instances per class.
>
> In **Appendix D.7, Table 11**, we provide results for CIFAR-10 comparing ASD with our method across all considered attacks.
> Although ASD achieves higher robustness than many existing defenses, our method remains consistently effective in removing poisoned samples and yields higher ACC with minimal ASR in every scenario.
> These results demonstrate that even under the relaxed assumption used by ASD, our approach delivers strong and superior defensive performance.
>
> **W4: Hyperparameters requires robustness testing.**
>
> In **Figure 8** and **Appendix Section D.2**, we investigate the impact of four major hyper-parameters used during coreset selection:
> label smoothing factor $\epsilon$, warm-up length $T_{warm}$, selection epochs $T_{se}$ and the regularization factor of unlearning $\gamma$.
> Despite the variance across different attacks, there exists a value range of applicability for each hyper-parameter.
> Using a value within the range (as used in our method) can ensure the reliability of our defense against different attacks
>
> **W5: Coreset might discard rare/complex case**
>
> Discarding samples might be challenging in preserving rare or complex cases.
> This impact oftentimes is measured by the class-wise accuracy [5,6].
> In Section 5.2 and Figure 6, we thus show the accuracy distribution across all classes.
> In comparison to the model after training on the entire dataset, the training on selected coresets from different poisoned datasets results in a closely the same class-wise accuracy distribution.
> Despite the ground-truth imbalanced accuracy, our method yields coresets that successfully preserve the fidelity of the original dataset in terms of the class-wise performance.
>
> Additionally, we visualize the class-wise accuracy distribution of experiments on GTSRB in **Appendix D.1, Figure 11**.
> Note that GTSRB intrinsically has a non-uniform (i.e., *imbalanced*) data quantity distribution across its 43 classes.
> In comparison to the accuracy distribution achieves when training on a fully clean dataset, our method consistently extracts a coreset that enables training a model with very similar class-wise accuracy distribution---even under a poisoning attack.

---

> ### Author Response · Authors · 2025-11-20
> **Reply to reviewer ERGX (2)**
>
> **Q1: What is the theoretical motivation of using min-max normalization?**
>
> During training procedure, the model outputs higher uncertainty for training samples in the early training stage.
> Thus, the naive application of the entropy will make the final CENT criterion heavily rely on several initial training epochs rather the entire training procedure.
> Motivated by this common yet important learning behavior, we use the min-max normalization on the entropy to avoid such a discrepancy over different epochs.
>
> In **Appendix Table 6**, we experimentally prove the positive effect of the min-max normalization with two example attacks, i.e., Blend and WaNet.
> Note that the latter learns the backdoor slower due to its original appending of noisy samples.
> Unlike Blend attack, using the min-max normalization has a noticeable improvement for the defense against WaNet, in particular at elimination of poisoned samples with a selection ratio of 0.5.
> In addition, balancing the entropy value by the min-max normalization can fairly reflect importance of samples contributing to late training time, which is beneficial for the accuracy preservation, which can be seen in Appendix Table 6 as well.
>
>
> **Q2: Is CENT similar to AUC calculation?**
>
> We thank the reviewer for this question.
> Mathematically, our aggregated entropy (CENT) essentially is equivalent to computing the area under the curve of each sample’s "normalized entropy vs. epoch" trajectory or computing the integral for that matter.
> In CENT, we accumulate the normalized entropy of each sample across training epochs, which corresponds to the AUC computed over this entropy curve.
>
> The distinction is therefore interpretational rather than mathematical.
> We use the cumulative entropy to emphasize learning dynamics and uncertainty evolution of each single sample, rather than as a classifier-evaluation metric.
> Thereby, the CENT criterion enables our coreset selection mechanism to effectively extract an informative and backdoor-free coreset.
>
>
> **Q3: How is the tail/edge-class accuracy?**
>
> To evaluate whether our coreset preserves performance on tail or edge-case classes, we report class-wise accuracy for models trained on our coreset vs. the full dataset in Table 6.
> As shown in Table 6(a), accuracy is nearly identical across all classes, including minority and tail classes, e.g. class 1 and 8 (cf. Table 6(b)), which confirms that our coreset selection does not disproportionately exclude hard or rare samples.
>
> Taking GTSRB as the example with an imbalance class-wise quantity distribution, ABCS results in a coreset that consistently achieves a similar class-wise accuracy distribution as the one after training on a fully clean GTSRB (see **Appendix D.1 and Figure 11**), showing the capability of preserving the dataset informativeness.
>
> ---
> ### References
>
> [1] Natarajan et al., "Learning with noisy labels," NeurIPS 2013.
>
> [2] Huang et al., "Uncertainty-aware learning against label noise on imbalanced datasets," AAAI 2022.
>
> [3] Kohler et al., "Uncertainty based detection and relabeling of noisy image labels," CVPR-W 2019.
>
> [4] Gao et al., "Backdoor Defense via Adaptively Splitting Poisoned Dataset," CVPR 2023.
>
> [5] Wei et al., "CFA: Class-wise Calibrated Fair Adversarial Training," CVPR 2023.
>
> [6] Wang et al., "Symmetric Cross Entropy for Robust Learning with Noisy Labels," ICCV 2019.

---

### Official Review · Reviewer_NEQB · 2025-10-25

**Soundness:** 3
**Presentation:** 3
**Contribution:** 3
**Rating:** 6
**Confidence:** 4

**Summary:**

This paper proposes a training-time backdoor defense method. Unlike traditional defenses, such as the model pruning or poisoned samples detection, this method adopts a coreset selection strategy to extract a subset of representative and clean samples from a poisoned training dataset. After this, the selected coreset is used to retrain the model, effectively addressing the impact of the backdoor attack. The key innovation of this work is the coreset selection criterion: the authors introduce a Cumulative Entropy (CENT)-based sample selection method that, through the Warm-up, Unlearning, and Label Smoothing stages, enables more accurate identification of clean samples, resulting in a high-quality and clean training subset.

**Strengths:**

1.	The paper introduces an innovative coreset selection method by proposing cumulative entropy as a metric to quantify the information content and uncertainty of samples, demonstrating a certain level of originality.
2.	The defense method does not rely on any additional clean datasets during implementation, and experiments conducted on multiple datasets under different attack scenarios show that ABCS significantly reduces the attack success rate.
3.	A lot of ablation studies is designed to directly reveal the influence of each parameter on the model.
4.	The method does not rely on any additional clean reference dataset, and its training time is comparable to standard training, demonstrating strong practicality.

**Weaknesses:**

1.	the proposed method lacks theoretical justification and instead relies on empirical statistics showing that poisoned and clean samples exhibit significant differences in the Cumulative Entropy metric.
2.	The method selects a coreset based on this metric, which reduces the overall information contained in the dataset; when the sample size is small, this reduction may affect the model’s primary accuracy.
3.	CENT prioritizes “high-information” samples, but this may introduce bias against representative samples from minority or rare distributions, thereby affecting fairness and generalization.

**Questions:**

1.	The proposed approach may reduce the training accuracy on datasets with limited sample sizes due to the smaller effective data volume. Please test the method on small datasets to confirm its stability.
2.	The paper relies on differences in cumulative entropy to distinguish poisoned from clean samples, but lacks theoretical justification or clear visual evidence for this assumption; providing a formal analysis or illustrative results would clarify why cumulative entropy effectively captures poisoning characteristics and strengthen the method’s credibility.
3.	Prioritizing high-information samples may bias against minority or rare data, reducing fairness and generalization; incorporating distribution-aware sampling or fairness regularization could help balance information gain and diversity.

---

> ### Author Response · Authors · 2025-11-20
> **Reply to reviewer NEQB**
>
> We thank the reviewer for this valuable feedback.
> While we do not include a theoretical framework, ABCS is grounded in extensive empirical observations across diverse attack scenarios.
> **Figure 4 and 5** show that the Cumulative Entropy (CENT) is highly effective at excluding the majority of poisoned samples.
> Additionally, we introduce a novel unlearning step integrated into each training epoch during coreset selection, which increases the uncertainty of hard yet informative samples, thereby amplifying the entropy gap between the benign and poisoned samples.
>
> Below, address the mentioned weakness and answer all questions:
>
> **W2/Q1: The possible primary accuracy reduction with limited data volume**
>
> In Appendix D.1, we additionally evaluate with the GTSRB dataset.
> In comparison to CIFAR10, GTSRB has fewer samples (39209 vs 50000) at a higher number of classes (43 vs 10) and thus significantly less samples per class.
> As shown in Table 5, our defense is robust against various attacks.
>
> While the dataset size/volume relates to effective anti-backdoor coreset selection, the informativeness of the dataset is more important than the mere number of samples.
>
> First, we note that ABCS is designed to not require a perfectly accurate dataset partition.
> This stands in contrast to identifying a globally optimal coreset size that simultaneously guarantees natural accuracy and maximal compactness.
> ABCS naturally filters out (a) poisoned samples and (b) benign but less informative samples. This allows preserving the model's natural accuracy while mitigating poisoning-based backdoor attacks.
>
> Second, ABCS provides an adaptive mechanism to identify the coreset size: mis-predicted samples are treated as hard-to-learn data and are excluded from the threshold computation (see lines [245–248]).
> This mechanism ensures that the selected coreset remains sufficiently large.
> Moreover, the automatically determined coreset sizes scale with dataset difficulty (**Appendix, Table 6**) and allow to finally train a backdoor-free model with high natural performance.
> Overall, the adaptive adjustments is beneficial over a fixed coreset size (Figure 5).
>
>
> **W3/Q3: Investigation of fairness and generalization using tail/edge-accuracy**
>
> Thank you very much for this question.
> We spent some time to additionally evaluate the class-wise accuracy across four different backdooring attacks and compare them with the model after training on the entire clean dataset. Results are summarized in Figure 6.
> Despite the non-uniform sampling across classes, training with ABCS yields a class-wise accuracy similarily distributed to training on the entire clean dataset.
>
> Incorporating fairness or generalization would be beneficial.
> However, in practice, the defender only received a poisoned dataset, for which the ground-truth "fairness" or generaliability is unknown---these properties might be altered by the poisoning.
> A native reinforcement still cannot ensure the fidelity similar to the original clean dataset.
>
> More challenging are intrinsically imbalanced dataset.
> Thus, we thus additionally visualize the class-wise accuracy on GTSRB, which has non-uniform distribution of samples across its 43 classes.
> As illustrated in **Appendix D.1 and Figure 11**, our method always selects a coreset that yields a similar class-wise accuracy distribution.

---

### Official Review · Reviewer_tyJb · 2025-10-26

**Soundness:** 2
**Presentation:** 3
**Contribution:** 2
**Rating:** 4
**Confidence:** 3

**Summary:**

The paper reframes training-time backdoor defense as coreset selection. It proposes ABCS, which warms up a model, ranks samples by cumulative entropy across epochs, performs a light unlearning step to widen the uncertainty gap between clean and poisoned data, then retrains from scratch on the top informative subset. It’s evaluated on CIFAR-10 and Tiny-ImageNet against multiple attacks (BadNets, Blend, CLB, IAB, WaNet, ISSBA, Low-Frequency, Adaptive Blend) plus an all-to-all variant. ABCS selects ~55% of data with near-zero poison rate and preserves clean accuracy while matching or improving total training time versus naive training.

**Strengths:**

1.Recasts training-time backdoor defense as coreset selection with a principled cumulative uncertainty criterion, rather than explicit poisoned/benign splitting.

2.Provides clear motivation that poisoned samples tend to be less informative (low entropy) while clean, hard examples remain high-entropy.

3.The CENT formula, normalization, and rationale are precisely specified. Figures contrasting epoch-wise vs. cumulative selection help understanding.

**Weaknesses:**

1.Given that the paper fixes the target label and uses a global poisoning rate, does this imply that poisoned samples are drawn from the entire training set and can originate from any source class? However, real attackers often poison non-uniformly (e.g., only a single source classes) and at low rates. It’s unclear how sensitive ABCS is to class-selective or distribution-shifted poisoning, where high-entropy informative clean samples may be concentrated in specific regions.

2.The work lacks analysis in terms of the selection-ratio of the coresets. The size is driven by an automatic thresholding heuristic on cumulative entropy, and in different settings the required ratio varies widely, leaving practitioners without bounds that relate coreset size to poisoning rate/attack strength or guidance on how small a coreset remains safe across datasets and threats.

3.While the paper mentions adaptive scenarios, there is limited exploration of attackers who maximize entropy of poisoned samples (e.g., trigger randomization, training-time augmentations designed to keep poisoned examples uncertain) to subvert ABCS.

4.The method is strongly dependent on hyperparameter choices (e.g., warm-up length, number of selection epochs, and unlearning strength), which may limit its plug-and-play usability and robustness in real-world deployments.

**Questions:**

Please refer to “Weaknesses”.

---

> ### Author Response · Authors · 2025-11-20
> **Reply to reviewers tyJb (1)**
>
> **W1: Lacking evaluation on single-source-class poisoning**:
>
> We thank the reviewer for this insightful comment. In our main evaluation, we follow the standard experimental setting as used in prior work, which assumes random uniform sampling of poisoned examples from the training set.
>
> However, we spent the last few days on study the sensitivity of ABCS in this regard.
> We first consider a knowledgeable attacker who performs an importance ranking using the CENT criterion and selectively poisons samples with high uncertainty.
> As reported in **Appendix D.8, Table 12**, this indeed impacts our defense.
>
> Second, we explore the robustness against a distribution-shifted poisoning.
> For this, we evaluate a *class-selective* strategy in which the attacker poisons only a single source class and simultaneously lowers the global poisoning rate to 1%, ensuring that most samples from that source class remain.
> The experimental results are visualized in **Appendix D.8, Figure 15**.
> While this scenario introduces higher variance in ASR, ABCS consistently maintains an average ASR below 5%.
>
> **W2: Lack of analysis of the automatic selection-ratio**:
>
> We appreciate the reviewer’s emphasis on our automated selection of the coreset size.
> Please note, that coreset selection (and thus the chosen size/selection ratio) is a mean to an end rather than an objective for itself.
> That is, we use coreset selection for backdoor removal not for dataset reduction in general.
> Consequently, there is no trade-off between dataset reduction and poisoning rate/attack strength or bounds for how far we can reduce the dataset to maintain backdoor resistance.
>
> As such the coreset size used by ABCS is dependent on the dataset (**Table 7 in Appendix**) and the backdoor/poisoning attack (Table 4).
>
> **W3: No exploration of adaptive attack that enlarges uncertainty of poisoned samples**:
>
> *1. Training augmentations*
> We intentionally disable data augmentations (e.g., Random Crop, Horizontal Flip) for coreset selection to avoid learning instability and variance in the uncertainty estimation.
> This ensures that our selection process reflects the intrinsic learning dynamics rather than augmentation-induced noise.
> Moreover, disabling the data augmentation helps stalibize the backdoor learning [1], enabling a more reliable exclusion of poisoned samples.
>
> *2. Adaptive adversary enhancing the uncertainty of backdoor*
> An adaptive adversary may aim to increase the learning difficulty or prediction uncertainty of poisoned samples.
> We additionally study the robustness of ABCS under two such attacks:
>
> (a) Trigger randomization attack.
> We we select four candidate triggers from BadNets, Blend, IAB and WaNet, and we randomly assign one trigger to each poisoned sample during data poisoning.
> In **Appendix Figure 16**, we can see a slower convergence (i.e., $\Delta H$ around zero) to the trigger randomization backdoor than a single Blend attack.
> Nevertheless, our defense remains a high robustness against the trigger randomization attack, finally resulting in a high ACC and consistenlty a minimal ASR for each involved attack, as shown in **Appendix Table 13**.
>
> (b) Label randomization for poisoned samples.
> In randomizing labels of poisoned samples, some poisoned samples receive randomized labels instead of the backdoor target.
> Such "label noise" has been shown to make training more stochastic and increase predictive uncertainty [2, 3].
> We evaluate this scenario across various backdoor attacks in **Appendix Table 14**.
> Label noise raises the learning difficulty for the backdoor, slows convergence, and increases prediction uncertainty, making it harder to separate benign from poisoned samples (cf. **Appendix Figure 17**).
> For instance, when 50% of poisoned samples are mislabeled, ASR drops to 40-50%, and as low as 20% for Adaptive Blend.
> Most Importantly, however, the effectiveness of our defense scales with attack strength.
> At high label-noise ratio (e.g., 50%), which weakens the backdoor itself, our defense still provides notable ASR reduction (e.g., WaNet ASR drops from 78% to 9%).
> At lower noise ratios, where the backdoor remains strong, our method achieves even larger ASR reductions, demonstrating robust performance across varying attack intensities.

---

> ### Author Response · Authors · 2025-11-20
> **Reply to reviewer tyJb (2)**
>
> **W4: Dependency on hyper-parameters**
>
> Apart from the analysis of label-smoothing factor $\epsilon$ in Figure 8, for the rebuttal, we have analyzed the impact of all other key hyperparameters in **Appendix Section D.2**, including the warm-up length ($T_{warm}$), selection epochs ($T_{se}$), and the regularization factor ($\gamma$).
>
> The unlearning strength is primarily controlled by $\gamma$, while the unlearning duration is fixed to one epoch after each training epoch during the selection stage.
> This design ensures that we consistently push intermediate samples with high uncertainty further away from easy-to-learn samples (including poisoned samples), thereby maintaining a better separation between benign and poisoned data.
>
> Although optimal configurations may vary slightly across different attacks, we find that each hyperparameter shows a stable operating range rather than requiring precise tuning.
> Parameters as used in our experiments are within this range and, thus, consistently ensure robust and reliable defense performance across diverse attack types.
> Consequently, our method maintains strong generalizability and practical usability, even under non-optimal hyperparameter settings.
>
> ---
>
> ### References
>
> [1] Qiu et al., "DeepSweep: An Evaluation Framework for Mitigating DNN Backdoor Attacks Using Data Augmentation," AsiaCCS 2021.
>
> [2] Huang et al., "Uncertainty-aware learning against label noise on imbalanced datasets," AAAI 2022b.
>
> [3] Kohler et al., "Uncertainty based detection and relabeling of noisy image labels," CVPR-W 2019.

---

> > ### Comment · Reviewer_tyJb · 2025-11-28
> >
> > I appreciate the authors’ detailed response and the additional experiments. Regarding W2, I find the explanation that “coreset selection is a means to an end” to be insufficient without further empirical evidence concerning the distributional properties of the selected coreset. Specifically, it remains unclear whether the Coreset maintains the same coverage as the original clean training set? To fully validate the safety and effectiveness of the coreset selection method, it would be highly beneficial if the authors could provide a visualization (e.g., t-SNE of the feature space) or a quantitative analysis that illustrates the distribution of the subsets.

---

> ### Author Response · Authors · 2025-11-29
> **Response to the evidence of distributional properties**
>
> Thank you for valuable suggestion to assess the distributional properties of the selected coresets.
> In latest revision (cf. **Appendix D.9**), we have incorporated both a feature-space visualization and additionally use two quantitative metrics to investigate whether the yield coresets maintain coverage of the original clean training set.
>
> **1. Visual Evidence via t-SNE**
>
> As shown in **Figure 18**, we use a ResNet18 pretrained on the full clean dataset and visualize the feature-space distributions based on the penultimate-layer representations for the full clean training set and all selected coresets from either the clean or poisoned training datasets.
> Across all settings, the coresets exhibit a high geometric similarity to the distribution of the full clean dataset, demonstrating that ABCS consistently preserves the global structure of the clean data manifold and adapts robustly across different conditions.
>
> **2. Quantitative Evaluation of Coreset Coverage**
>
> In addition to the t-SNE plots, we provide two quantitative metrics to evaluate how well each coreset covers the full clean dataset:
> (1) Local geometric coverage---measured by the 95th-percentile nearest-neighbour distance (*p95-NN*) using cosine distance.
> (2) Global distributional similarity---measured by Maximum Mean Discrepancy (*MMD*) with an RBF kernel.
> Both metrics attain a value of \num{0} only when achieving full coverage or an identical distribution, respectively.
>
> **Table 15 in the Appendix** summarizes the results.
> When comparing each coreset to the full clean dataset, both *p95-NN* and *MMD* remain below 0.015, indicating a strong local coverage and high global similarity.
> Some class-wise variance naturally appears, though:
> easy-to-learn classes (e.g., classes 1 and 8) can tolerate small dissimilarity to the full data distribution, whereas difficult classes (e.g., class 3 and 5) benefit from a larger selection and consequently exhibit extremely low *p95-NN* and *MMD* values, ensuring preserving natural performance.

---

### Official Review · Reviewer_e9nn · 2025-11-01

**Soundness:** 3
**Presentation:** 3
**Contribution:** 3
**Rating:** 6
**Confidence:** 3

**Summary:**

This paper proposes a novel training-time defense against neural backdoor attacks called Anti-Backdoor Coreset Selection. The core idea formulates the defense as a coreset selection problem, aiming to extract a small, informative, and benign subset from a poisoned training set. The authors introduce the Cumulative Entropy criterion, which accumulates the prediction entropy of each sample over multiple training epochs to identify data valuable for the primary task and unlikely to be poisonous. ABCS consists of three phases: a warm-up phase, a coreset selection phase, and a final training phase on the selected clean coreset. Experiments on image and text benchmarks demonstrate that ABCS effectively mitigates various backdoor attacks, maintains natural accuracy comparable to training on a fully clean dataset, and incurs computational costs similar to standard training.

**Strengths:**

1. This paper reframes the backdoor defense problem during training as a coreset selection problem, breaking from the traditional dataset splitting approach. By leveraging the coreset concept from data-efficient learning for defense, it provides a novel and theoretically grounded perspective for training-time backdoor defense. This design is clearly demonstrated through the definition of ABCS and its comparison with traditional defenses.

2. The proposed Cumulative Entropy criterion effectively distinguishes benign from poisonous samples by accumulating entropy values over multiple epochs. In experiments, it performs best in reducing the poisoning rate of the coreset and the Attack Success Rate, forming the core support for ABCS's defensive effectiveness.

3. It does not rely on a clean reference dataset, avoiding manual vetting costs. The coreset size is smaller than the full dataset, leading to faster final training. The overall runtime is comparable to naive training, making it suitable for practical application scenarios.

**Weaknesses:**

1. The paper only empirically validates the effectiveness of ABCS without providing a theoretical explanation for why cumulative entropy separates benign and poisonous sample distributions better than single-epoch metrics or the mathematical relationship between coreset selection and backdoor defense effectiveness. The conclusion section of the document explicitly acknowledges this limitation.

2. Against adaptive attacks where the adversary knows the CENT criterion, ABCS's ASR increases significantly (ASR rises to 11.69% for A-Blend and 6.44% for WaNet), indicating a shortcoming in defending against highly knowledgeable attackers.

3. The selected coreset size is larger than the optimal coreset for a clean dataset, and the paper lacks a deep analysis of the trade-off between coreset size and defensive robustness/natural accuracy. There is potential for further data size reduction.

**Questions:**

1. Why is extracting a partial benign coreset superior to splitting out all benign samples, especially at low poisoning rates? Please support the necessity of this framework choice.

2. In Table 4, the coreset selection ratio $r_{se}$ for the WaNet attack (57.83%) is higher than for the Blend attack (53.85%). Is this difference directly related to the impact of these two attack types on sample learning difficulty? Furthermore, for stealthier attacks like A-Blend, is it necessary to adjust the currently fixed label smoothing factor $\epsilon$ (0.9) to enhance the unlearning effect, or is this $\epsilon$ value universally applicable across attack scenarios？

---

> ### Author Response · Authors · 2025-11-20
> **Reply to reviewer e9nn**
>
> We sincerely appreciate the reviewer's feedback and acknowledge the weaknesses pointed out.
> Our method is grounded in a practical understanding of learning dynamics under poisoning and is validated by consistent experimental performance.
> In this scope, we also extensively evaluated adaptive attackers that indeed shown an impact on defense performance---however, **significantly less than for related work**.
> Investigating adaptive attacks is crucial for evaluation of defenses.
>
> The third weakness on the coreset size, we would like to answer alongside the other questions in more detail below:
>
> **W3: Selected coreset size is larger than optimal size**:
>
> The selected coreset size in our method is larger than the optimal size typically used for clean datasets.
> However, the goal of our approach also is fundamentally different:
> rather than striving for a minimal yet perfectly representative coreset, we focus on extracting an informative and backdoor-free subset that enables robust model training.
> Unlike traditional coreset methods that rely on fixed sampling ratios, our approach thus determines the coreset size via an adaptive threshold based solely on correctly predicted samples.
> As shown in **Appendix Table 7**, this adaptive threshold naturally yields a larger coreset size.
> Consequently, there indeed exist room for further improvement from a "data reduction perspective" that can complement backdoor suppression.
> We leave this complementary objective to future work.
>
>
> **Q1: Why extracting benign coreset is superior to splitting out all benign samples?**
>
> The extraction of a coreset allows split informative, backdoor-free samples and leave easy-to-learn benign samples and poisoned samples behind as a "tolerated compromise".
> Easy-to-learn benign samples and poisoned samples are difficult to tell apart.
> Essentially, we avoid the "gray area" where the model might have higher prediction uncertainty on poisoned samples than few easy benign samples.
> Thus, a precise and complete separation actually brings a higher chance of mis-splitting, which often leads to a defense failure.
> In a way, we favor recall over precision allowing for small portion of false positives, that are, the easy-to-learn benign samples.
>
> Lowering the poisoning rate will preserve more benign samples in the training dataset. Nonetheless, a successful backdoor attack should achieve both high ASR and maintain a comparable ACC.
> Thus, a lower poisoning rate actually does not change the major information contained in the dataset, which the coreset should cover.
> However, as shown in **Appendix D.3, Figure 13** under Blend attack, learning on a dataset with low poisoning rate actually exhibits even a higher loss values on poisoned samples due to the lower appearance possibility in each training batch.
> In comparison to defenses explictly relying on dataset splitting, the increased confusion between benign and poisoned samples in the view of prediction losses at early training epochs will make the splitting more challenging.
> Thus, selecting a coreset can avoid the high risk of involving poisoned samples into the final training set, achieving a robust and reliable defense.
>
> **Q2a: Variant coreset size across different attacks**
>
> The coreset selection ratio naturally varies across attack types.
> For WaNet and Adaptive-Blend, the selected ratios are slightly higher than those for other attacks, which is consistent with the naive training results presented in Table 1(a).
> Both attacks yield slightly lower test accuracies, indicating larger learning difficulty and more frequent mispredictions.
> As a result, our adaptive threshold selects a larger coreset to ensure sufficient coverage of these more challenging learning dynamics.
> This behavior reflects an important property of our method: the coreset size adapts automatically to the intrinsic difficulty of the poisoned dataset, rather than being attack-specific or manually tuned.
>
> **Q2b: Use of label smoothing factor $\epsilon$**
>
> We adhere to our threat model in which the defender has no prior knowledge of the attack existence or its type.
> Thus, tuning $\epsilon$ based on one attack would violate the assumptions of the setting.
> We therefore use a fixed value of $\epsilon = 0.9$ for all CIFAR-10 experiments to maintain fairness and general applicability.
> Empirically, increasing $\epsilon$ strengthens the unlearning effect (cf. Figure 8), and $\epsilon = 0.9$ already yields a strong unlearning capability.
> This makes it a robust default choice even for stealthier attacks such as Adaptive-Blend attack.
>
> ---
>
> ### References
>
> [1] Paul et al., "Deep learning on a data diet: finding important examples early in training," NeurIPS 2021.

---

> > ### Comment · Reviewer_e9nn · 2025-11-25
> >
> > I appreciate the authors’ careful and comprehensive responses to my comments. I now have a clearer understanding of the proposed method and do not have any further questions. Since I have already given a relatively positive score, I will keep our score unchanged.

---

### Author Response · Authors · 2025-11-20
**Revision Summary**

We sincerely thank to all reviewers for their valuable comments and insightful questions. In our revision (cf. text highlighted in blue color), we have incorporated all experiments and explanations following reviewer's suggestions.
In particular, we:

**Appendix D.1** (Reviewers: NEQB-W3/Q3, ERGX-W5/Q3)

* Add the class-wise accuracy distribution on GTSRB (cf. Figure 11) as a showcase for imbalanced dataset to demonstrate ABCS's ability to preserve dataset informativeness in terms of fairness and the tail/edge-accuracy.

**Appendix D.3** (Reviewer: e9nn-Q1)

* Visualize the learning procedures under different poisoning rates (cf. Figure 14), illustrating that lowering the poisoning rate increases the difficulty of dataset splitting.

**Appendix D.7** (Reviewer: ERGX-W3)

* Add experimental comparison to a defense that uses a clean data pool as reference (ASD; Gao et al., CVPR 2023). Results are summarized in Table 11.

**Appendix D.8** (Reviewers: tyJb-W1, tyJb-W3, ERGX-W2)

* Investigate a class-selective and distribution-shifted adaptive adversary that poison a single non-target class only AND lowers the global poisoning rate at the same time (cf. Figure 15);

* Evaluate on a trigger randomization attack that increases the predictive uncertainty of poisoned samples, thereby slowing convergence to them, by randomly assigning a trigger pattern to each poisoned sample (cf. Figure 16 and Table 13);

* Use label randomization for poisoned samples to model an adversary that enforces slow convergence and uncertainty maximization during backdoor learning (cf. Figure 17), and we summarize corresponding experiments in Table 14.

---
### Further revision after the discussion

After the discussion phase with reviewers, we:

**Appendix D.9** (Reviewers: tyJb-W2)

* Provide visual evidence using t-SNE to show that selected coresets---whether from clean or poisoned datasets---consistently retain high coverage of the full clean dataset (cf. Figure 18);

* Quantitatively assess the coreset coverage relative to the full clean training set using (a) the 95th-percentile nearest-neighbor distance (p95-NN) to measure local geometric coverage and (b) the maximum mean discrepancy (MMD) to evaluate global distribution similarity. Both assessments, applied to either individual classes or the full coreset, demonstrate strong coreset coverage of the full clean dataset (cf. Table 15).

---

### Comment · Area_Chair_zGe4 · 2025-11-22

Dear reviewers,
Please check the authors’ responses. As there are differing opinions about the paper, it would be appreciated if you could evaluate—based on all comments—whether the authors have adequately addressed the main concerns.

---

### Author Response · Authors · 2025-12-03
**Global response to reviewers and chairs**

Dear Reviewers, ACs, SACs, and PCs,

We sincerely thank you once again for your time, effort, and insightful feedback on our submission. Your constructive comments have been invaluable in helping us improve the manuscript.

Over the past few days, we have carefully revised the manuscript in response to all review comments and our further discussions with the reviewers. A summary of the revisions can be found in our prior reply here: [*Revision Summary*](https://openreview.net/forum?id=7Izu4XMELq&noteId=8Jn0wVh4YW).

All changes in the revised manuscript are highlighted in blue for ease of reference. We hope that our detailed rebuttal and the updated manuscript clarify and strengthen the contributions of our work and facilitate a smooth evaluation of our submission.

---

### Meta-Review · Area_Chair_dqof · 2025-12-22

**Summary:**

Reviewers agreed that the paper is well-motivated and presents a clear and effective training-time backdoor defense by reframing the problem as coreset selection and using cumulative entropy to identify informative, benign samples. The method is simple, does not require clean reference data, and shows strong empirical performance across multiple datasets and attacks with minimal impact on natural accuracy.

However, reviewers also raised concerns about the limited conceptual novelty, as the core components—uncertainty/entropy-based selection, accumulation over epochs, and learning-dynamics signals—are closely related to prior work. In addition, the approach relies on several heuristic design choices and lacks theoretical justification. While the rebuttal added experiments and clarified robustness to several attacks, concerns remain regarding adaptive threat models and whether the selected coreset is optimal or fundamentally superior to dataset splitting.

Overall, the rebuttal addressed most empirical and clarification issues, but questions about novelty depth and theoretical grounding remained central in shaping the recommendation.

**Reviewer Concerns:**

Concerns addressed by the rebuttal:
The rebuttal adequately addressed several reviewer concerns by adding clarifications and additional experiments. In particular, it clarified the motivation for cumulative entropy over single-epoch uncertainty measures, provided further empirical analysis on coreset properties (e.g., class-wise distribution and coverage), and included additional results on different datasets and attack settings. Reviewers’ questions regarding implementation details, hyperparameter choices, and stability were largely resolved, and no major soundness issues remain.

Concerns still outstanding:
Some concerns remain partially unresolved. Reviewers continued to question the depth of novelty, as the method largely combines existing ideas such as uncertainty-based selection, accumulation over epochs, and learning-dynamics signals. The approach also relies on several heuristic design choices (e.g., warm-up, normalization, unlearning via label smoothing) without strong theoretical justification. In addition, while robustness to standard attacks is demonstrated, concerns about strongly adaptive or entropy-aware attackers and the fundamental optimality of coreset selection over dataset splitting were not fully addressed.

**Reviewer Scores:**

Reviewer e9nn:
This reviewer explicitly stated that the rebuttal addressed their questions and that they would keep their score unchanged. I therefore expect no change.

Reviewer tyjb:
The rebuttal addressed several technical concerns with additional experiments and clarifications, but did not fundamentally change the reviewer’s concerns regarding novelty and heuristic design. I expect the score to be unchanged or at most a very slight increase.

Reviewer ERGX:
While the rebuttal improved clarity and provided further justification for design choices, the reviewer’s core concerns about novelty and assumptions behind cumulative entropy likely remain. I expect the score to be unchanged.

Reviewer NEQB:
This reviewer viewed the paper positively, highlighting the empirical effectiveness and clarity of the proposed cumulative entropy criterion, while raising concerns mainly about potential bias, theoretical grounding, and broader generalization. The rebuttal addressed several of these points with additional experiments and clarifications, but did not fundamentally alter the conceptual assessment. I therefore expect the reviewer’s score to be largely unchanged, or at most to show a very slight positive adjustment.

---

### Decision · Program_Chairs · 2026-01-26

Reject